# Ultrasound Systems for Biometric Recognition

**DOI:** 10.3390/s19102317

**Published:** 2019-05-20

**Authors:** Antonio Iula

**Affiliations:** School of Engineering, University of Basilicata, 85100 Potenza, Italy; antonio.iula@unibas.it

**Keywords:** biometrics, ultrasound imaging, ultrasonic transducers

## Abstract

Biometric recognition systems are finding applications in more and more civilian fields because they proved to be reliable and accurate. Among the other technologies, ultrasound has the main merit of acquiring 3D images, which allows it to provide more distinctive features and gives it a high resistance to spoof attacks. This work reviews main research activities devoted to the study and development of ultrasound sensors and systems for biometric recognition purposes. Several transducer technologies and different ultrasound techniques have been experimented on for imaging biometric characteristics like fingerprints, hand vein pattern, palmprint, and hand geometry. In the paper, basic concepts on ultrasound imaging techniques and technologies are briefly recalled and, subsequently, research studies are classified according to the kind of technique used for collecting the ultrasound image. Overall, the overview demonstrates that ultrasound may compete with other technologies in the expanding market of biometrics, as the different commercial fingerprint sensors integrated in portable electronic devices like smartphones or tablets demonstrate.

## 1. Introduction

At first, biometric recognition systems were mainly exploited for applications in public security and criminal investigation but, more recently, they are being used in more and more civilian fields because they proved to be reliable and accurate. Several possible biometric characteristics can be used, including face, iris, fingerprint. The choice depends on the specific application; to this end several issues have to be taken into account: universality (all people should have the characteristic), uniqueness, permanence over a reasonable period of time, collectability, acceptability, performance and circumvention (should be resistant to spoof attacks) [1,2]. The human hand has been employed for extracting various biometric features like fingerprint, palmprint, hand geometry and hand vein pattern. Fingerprint has been used in criminal investigation since the early 1900s, and, nowadays, it is still one of the most popular biometrics [3,4,5]. A fingerprint contains information on ridge and valley patterns of a human fingertip. Depending on sensor resolution, there are different types of features that can be exploited in recognition processes [6]: at global level patterns like loop, whorl, and arch, at local level patterns like ridge endings and bifurcations called minutiae and, finally, at intra-ridge level patterns of sweet pores.

The palm of the human hand contains patterns of ridges and valleys similar to those of the finger [7,8,9] and, due to its larger area, palmprint should be even more distinctive than fingerprint, but slower. Another possibility is to acquire a palm image with a low resolution scanner; in this case, extracted features are principal lines and wrinkles only [10,11]. Low-resolution palmprint images are mainly used in access control applications.

Hand vein pattern is a biometric characteristic that is attracting growing attention from both researchers and companies [12,13]. Vascular patterns of the hand are very distinctive and stable over a long period of time [14]; also, they are very difficult to counterfeit because they cannot be seen by the human eye as they are hidden under the skin.

Hand geometry uses measurements of palm size, finger length and width as features [15]. Recognition systems based on hand geometry are quite simple and inexpensive but they are only suited for a relatively small population group.

For many years, biometric recognition systems only exploited 2D images and features, despite of the fact that human characteristics are 3D. More recently, several methods have been proposed for achieving 3D information from biometric characteristics like face, palmprint, fingerprint and ear [16].

A recognition system that uses only one biometric characteristic is known as unimodal while, if two or more independent characteristics are collected, the system is called multimodal. This last recognition modality is often used when there is the need to improve the recognition rate [17,18,19].

Several technologies have been used to acquire images of the various biometrics including optical, capacitive, thermal and infrared. The main merit of ultrasound over other technologies is the capability of collecting volumetric images of the investigated biometrics, which produces twofold benefits: the possibility to extract features that are more distinctive and a very strong resistance to spoof attacks. In addition, ultrasound images do not suffer from many kinds of surface contaminations, humidity or ambient light.

The present work reviews research activities devoted to develop ultrasound systems for biometric recognition purposes. Several sensor technologies, from classic piezoelectric or piezocomposite transducers to the most recent capacitive and piezoelectric micromachined ultrasonic transducers, CMUT and PMUT, respectively, as well as different imaging techniques have been experimented over the years. Fingerprint is the most investigated biometric characteristic, especially since it was clear that authentication in smartphone devices would have been done mainly through fingerprint. The possibility of collecting other biometrics including vein pattern, palmprint, hand geometry has been explored as well in the recent literature. In the present paper, research studies are classified according to the kind of ultrasound technique exploited for acquiring the image. Hereafter, Section 2 briefly recalls main ultrasound techniques and transducer technologies used for imaging the human body. Section 3 and Section 4 analyze research papers that used pulse-echo and impediography techniques, respectively. In Section 5, works that experimented with different approaches are presented and, finally, conclusions are drawn in Section 6.

## 2. Fundamentals of Ultrasonics

Even if piezoelectricity was discovered by Jacques and Pierre Curie in 1880, the first human application of ultrasound was due to the French physicist Paul Langevin who, after the Titanic sinking (1912), started experimental studies that brought to first sound navigation and ranging (SONAR) transducers based on quartz to detect icebergs and submarines [20,21,22].

Later on, a strong impulse to ultrasonic applications came with the discovery of strong and stable piezoelectric properties in ceramic materials like barium titanate, lead niobate and mostly lead zirconate titanate (PZT). These materials, being composed of solid mixtures of powders, are able to provide a wide range of operating parameters [23].

Nowadays, ultrasound is applied to several fields. For example, ultrasound imaging or sonography is widely employed in medicine and in nondestructive evaluation of products and structures, where it is used to reveal invisible flaws, but also to detect objects and measure distances. Other industrial applications of ultrasound include cleaning, mixing, and accelerating chemical processes.

### 2.1. Ultrasound Imaging Techniques

Ultrasonic images are a very useful tool in medical diagnostics and new techniques are continuously developed [24,25,26]. 3D ultrasound is used in several applications because it is able to provide more complete information on human anatomy than 2D ultrasound [27,28,29]. Main techniques able to provide ultrasonic images are briefly described in the following.

#### 2.1.1. Pulse-Echo Imaging

The basic modality of pulse-echo imaging is the amplitude (A)-mode: the wave transmitted by a transducer propagates through the body and reflected echoes provide information on the depth of tissue interfaces. The simplest way to obtain a two-dimensional image consists in moving the single transducer along a direction while acquiring several A-modes. After processing the A-line signals, a cross-sectional image is obtained and with some further processing a gray scale brightness (B)-mode image can be rendered. B-mode images are usually obtained by exploiting an array of transducers, which is faster and more effective because of the electronic scan and allows beamforming techniques like focusing, steering and apodization.

Similarly, three-dimensional ultrasound images can be obtained either using a single transducer and performing mechanical scans along two orthogonal directions or using a linear array and performing a single mechanical scan. Another possibility relies in the use of 2D transducer arrays, which however imply severe technological challenges [30,31].

A C-mode image is a 2D image orthogonal to a B-mode image, at a given depth from the transducer. Whenever a volumetric image is acquired, any B-mode or C-mode image can be easily extracted as shown in Figure 1.

Doppler ultrasound imaging is widely used in cardiovascular analysis. It exploits the Doppler effect and is able to image the movement of tissues or blood. Color Doppler images are often superimposed to B-mode images and provide information on the speed and direction of the motion in a color scale. Power Doppler ultrasound, instead, displays only blood flow, but with better detail than color Doppler [32,33].

#### 2.1.2. Impediography

The above-mentioned methods use the pulse-echo technique to provide images that map the reflectivity of scanned tissues. Another possibility to achieve acoustical images consists in evaluating the acoustic impedance of the load by means of a simple measurement of the electric impedance of the transducer. This technique is usually known as impediography: when the ultrasonic wave finds a change of acoustical impedance, part of the energy is transmitted and part is reflected [24]. The higher is the difference between the acoustic impedances of transducer and load, the higher is the reflected wave. Based on this simple principle, loads with different acoustic impedances can be distinguished by simply analyzing the electrical impedance of the transducer. As an example, Figure 2 shows the electric impedance of a single piezoceramic element, which has an acoustic impedance of around 30 MRayl, simulated by using a 1D model [20,23], when it is loaded by water (acoustic impedance about 1.5 MRayl) and air (acoustic impedance about 400 Rayl), respectively.

As can be seen, different loads can be easily detected by evaluating the minimum value of the electrical impedance modulus (which occurs approximately at the series resonance frequency $f_{s}$).

#### 2.1.3. Other Techniques

There are other techniques that can be exploited to produce ultrasound images, including photoacoustics and acoustic holography.

The photoacoustic (PA) effect allows the conversion of light to sound. When a tissue is exposed to high energy, like for example laser light, it absorbs part of this energy causing localized heating and rapid thermoelastic expansion that produces transient broadband ultrasound waves [34]. The magnitude of the ultrasonic signal directly depends on the local energy deposition; in this way, 2D or 3D images of the areas of interest can be formed [35].

Near-field acoustic holography (NAH) is a technique that is based on performing a measurement in the near field of the source with a number of transducers and then on reconstructing the whole three-dimensional sound field. NAH allows to estimate sound pressure, particle velocity, and sound intensity vectors in a different position than the one measured, achieving enhanced spatial resolution, because it also acquires evanescent waves in the near-field [36].

### 2.2. Transducer Technologies

To be used in medical diagnostic imaging, ultrasonic transducers should be able to generate and receive short pulses with high sensitivity and spatial resolution in the MHz frequency range.

Piezoceramic materials like PZT are good candidates because of their high electromechanical coupling factor. However, they have a high acoustic impedance (about 30 MRayl) while that of the human body is around 1.5 MRayl. Consequently, a direct application of a PZT element would produce a high impedance mismatch, resulting in a high quality factor Q [37]. In order to increase transducer bandwidth, two adjustments are usually adopted. The first one consists of using an adequate backing material, which absorbs a large part of the energy transmitted by the piezoelement. The second one consists in inserting at least one impedance matching layer at the front of the transducer. Several theoretical and experimental analysis aimed to optimize the design of the matching layer, depending on the application, can be found in the literature [38,39,40,41]. An acoustic lens is often added to enhance resolution. When applied to the human body, a coupling medium (usually water or gel) is interposed between transducer and skin in order to avoid air bubbles that would reflect the ultrasound wave.

Nowadays, transducers based on piezocomposite technology are those used in most of the applications, while the emerging technologies of micromachined ultrasonic transducers (MUTs) are demonstrating very attractive, mainly because of their capability to integrate transducer and electronics in only one chip.

#### 2.2.1. Piezocomposite Transducers

Piezocomposite material is an important update of bulk piezoceramic. Piezocomposite transducers were introduced in the early eighties of the past century [42]. These materials are made by combining piezoceramic elements with epoxy or other polymer materials. The percentage of PZT material is usually less than 30 %, which results in a drastic decrease of the whole acoustic impedance and hence in a better coupling with the human body. Most commonly piezocomposite transducers used in medical ultrasonic imaging are of the 1–3 and 2–2 connectivity [33].

With respect to classical piezoceramic materials, piezocomposites provide in general higher energy conversion efficiency and larger bandwidth. Despite of their well-established use, an intense research activity on this kind of transducer is still carried out [43,44,45].

#### 2.2.2. CMUT

A capacitive micromachined ultrasonic transducer (CMUT) is composed of a high number of cells vibrating in flexural mode [46,47]. A single cell is schematically described in Figure 3a. Basically, it is a small capacitor where one of the electrodes is deposed on a silicon nitride membrane, which is suspended on an air cavity, while the other one is in the silicon substrate. A bias voltage of more than 100 V has to be applied together with the alternate signal to produce the flexural vibration of the membrane. As CMUT has a very low mechanical impedance, it was first conceived for in-air applications [48], but subsequent in-water experiments demonstrated its suitability for medical diagnostic applications, mainly thanks to its wide frequency bandwidth [49,50]; hence, several different technological solutions for CMUT realization have been proposed during the years [51,52,53]. The possibility of integration of CMUT with CMOS technology has been demonstrated as well [54], even if the high bias voltage may limit integration in portable devices.

#### 2.2.3. PMUT

Piezoelectric micromachined ultrasonic transducers (PMUTs), similar to CMUTs, are based on a flexural vibration of a membrane (see Figure 3b) but, in this case, the membrane is composed of at least one piezoelectric layer and one passive elastic layer. The lateral strain generated from the first one is contrasted by the second one, resulting in a flexural motion [55,56,57,58]. A main merit of PMUT over CMUT is that PMUT doesn’t need for high biasing voltages, which allows for an easier integration into portable electronics. PMUT devices are realized by using two main piezoelectric materials: aluminum nitride (AlN) and PZT. The much higher piezoelectric constants of PZT makes it preferable whenever high performance devices have to be realized, while AlN MEMS devices are usually used when integration with electronics is the primary issue, due to their compatibility with the CMOS process [59,60]. An important drawback of the PMUT is the relatively low bandwidth; to solve this issue several solution are being investigated [57,61].

## 3. Biometric Systems Based on Pulse-Echo Imaging

First experiments devoted to acquire live scan fingerprint images through an ultrasound system were probably carried out in the early nineties of the past century at Niagara Technology Laboratories, Buffalo, New York (USA) [62,63]. As probe the authors used a single piezoelectric transducer, with a fixed focus and aperture size of 6.35 mm, working at 30 MHz and operating in pulse-echo modality. An external focusing lens with focal length of 25 mm allows the transducer to produce a spot size of about 0.2 mm. The finger is placed on a collimated lens made of polystyrene and acoustically coupled through water. The ultrasonic beam from the transducer strikes a rotating acoustic mirror and is redirect to the lens, which basically converts the scan geometry from B-mode to C-mode. The echoes are sent back to the transducer to be processed and stored by the electronics. The same operation is repeated while a linear actuator moves the whole assembly along a direction orthogonal to the scan line and acquires successive lines of the fingerprint. In this way a total area of 19 × 19 mm${}^{2}$ was scanned and the achieved plain fingerprint rendered. The system resulted cheap and fast if compared to commercial high performance and cost scanning systems of that period, yet the achieved image quality was not adequate because the focal length of the transducer was too short. Some years later, the research group proposed an acquisition system based on a tightly focused transducer (frequency 30 MHz, aperture about 7.4 mm, focal length 5.9 mm) that is swept across the whole fingerprint area: a torque actuator, attached to a high resolution optical encoder, moves the transducer along one direction, while a brushless DC stepper motor provides the scan along a perpendicular axis [63,64]. This approach, even if more complex, guaranteed a resolution of 500 dpi. Acquired images of fingers contaminated by ink demonstrated that, different from optical systems, ultrasound images are not affected by such kind of contamination.

The idea to reproduce an ultrasonic fingerprint image by means of mechanical scans along two orthogonal axes was resumed few years later [65]. In this case a focused transducer working at 50 MHz allowed to achieve a resolution of 1000 dpi. The finger is pressed against a polystyrene plate 2 mm thick; to achieve good acoustic coupling, a gel layer is put between plate and finger and, furthermore, images are acquired in basin filled with water. The A-scan signals are then collected and stored in a 3D matrix and subsequently rendered in the form of several 2D images as B-mode or C-mode images. These images allow to visualize sweet pores that provide additional information and that cannot be detected with optical techniques. The shortcoming of this method is the relatively long acquisition time.

To overcome this limit the acquisition speed was greatly increased by exploiting a cylindrical scanning in one of the two directions [66]. To this end, several focused transducers are mounted on a cylinder that rotates around the finger. Transducers are turned on only when they face the area containing the fingerprint, while a linear scan along the axis of the cylinder allows to acquire the volumetric image. Successively, with the same approach, the research group demonstrated the possibility to obtain fingerprint patterns from under-skin layers, which makes the system independent from natural or voluntary deterioration of skin surface [67].

Another approach to achieve 3D ultrasound images of fingerprint used a 192 elements linear array probe, based on CMUT technology [52], which was specially designed for near-field ultrasound imaging [68]. To this end, a very small pitch, i.e., the distance between two elements of the array, was chosen (112 μm) in order to have the focal region very close to the transducer (from 3 to 10 mm); an acoustic lens provides focusing in elevation at 6 mm. The transducer exhibits a wide bandwidth (about 100%) with center frequency of 12.5 MHz.

3D images of a fingertip were acquired by driving the probe with a commercial ultrasound imaging system (Technos-Esaote, Italy) and automatically moving it along the elevation direction. Water was used as coupling medium. Figure 4a shows a 3D representation of a collected fingerprint, while Figure 4b a 2D image extracted at an under-skin depth of 0.2 mm. In both images, sweat pores are clearly visible. The design of the array was subsequently optimized by means of a FE model [69], which accounts for the acoustic lens and the backing. The same set up was exploited to provide images of a region of the human palm as well, by using a similar CMUT probe but with a larger aperture [70].

A relevant research activity has been carried out to develop a fingerprint sensor based on a AlN PMUT 2D array integrated with portable device electronics. The first implementation consisted of a 24 × 8 array with pitch of 100 μm, for a total area of 2.3 × 0.7 mm${}^{2}$. The electronics was realized in 180 nm CMOS technology [71,72]. A plane wave is generated by driving all PMUT elements with a 28 V 2-cycle 22 MHz square-wave; in other experiments a 12 V 3-cycle 20 MHz square-waves is used instead. In both cases, one column is selected and the received signals from 8 elements are read out in parallel. The time employed to image a fingerprint area of 2.3 mm × 0.7 mm (same area of the sensor) was equal to 24 μs. A mechanical scanning was performed to image higher areas; in this way, an image of a 2D polydimethylsiloxane (PDMS) fingerprint phantom with pitch of about 600 μm was acquired by using a liquid coupling layer (Fluorinert). Also, the authors presented an experimental study where they demonstrate that a very thin layer of material, used for sensor protection, results transparent to acoustic reflection even if a high acoustic impedance mismatch occurs [73]. Subsequently, the same research group upgraded sensor’s design by increasing array’s size to 65 × 42 and then to 110 × 56 elements, arriving to acquire a fingerprint area of 4.73 × 3.25 mm${}^{2}$ [74,75,76,77,78,79]. The authors provided a full acoustic characterization of the device, which showed lateral and axial resolution of 75 μ and 150 μ, respectively, and an adequate contrast ratio between ridge and valley (5:1). Also this kind of sensor is able to collect fingerprint images at both the epidermis and the dermis layer. The overall characteristics of the device seem to be very close to the requirements of consumer electronics.

An example of PMUT based on PZT can be found in the literature as well [80]. The transducer is composed of an array of 50 × 50 PMUTs and is fabricated by using a sol-gel PZT technique. The size of the cells is 50 μm while the pitch is 100 μm. The device has a resonance frequency of about 25 MHz and exhibits very good electromechanical properties. However, even if it is claimed that the transducer was specially designed and simulated for fingerprint applications, no image has been provided yet.

A finite element method study investigated the feasibility of ultrasonic transducers based on 1–3 piezocomposite to detect fingerprint patterns trough pulse-echo methods [81].

As discussed in the introduction, the possibility of extracting more than one biometric characteristic from the same acquired volume, realizing in this way a multimodal biometric system, could improve the overall performances of the biometric system. Such approach was probably investigated for the first time for recognition purposes in [82]. In that paper, a commercial ultrasound system (GE Logiq-9 scanner with a 14 MHz GE M12L probe) was employed to image the inner structure of fingers to detect and measure anatomical elements as bone contour or tendon with the aim of using them as potential biometric identifiers. The 3D images were collected by moving the probe along the elevation direction and by acquiring B-mode images with an interval of 0.4 mm. Some verification experiments carried out on a small database have shown the potential of the proposed methods.

A similar ultrasound technique has been widely exploited to extract other biometric characteristics than fingerprint by acquiring a volumetric region of the human palm.

3D palm under-skin images were firstly presented in Refs. [83,84]. Also in those works, a commercial equipment was used (Technos ultrasound scanner with a 7.5 MHz LA523 probe, both from Esaote, Italy) and the acoustic coupling between probe and hand was realized through a gel pad; in this case, due to the larger investigated volume, a higher number of different human organs like tendons, muscles and veins were identified and measurements of their dimension, together with some distances among them, were proposed as a biometric template. In subsequent works, the same research group experimented with the same approach with the aim to extract 3D palmprint images [85,86]. With respect to previous experiments, a 12 MHz frequency probe (LA435, Esaote) was used; furthermore, the acoustic coupling was realized through water. Results have shown that two kinds of 3D information can be extracted: the 3D profile of the palm, which can be achieved with optical methods based on structured-light imaging as well [87], and palm lines’ depth, which instead can be achieved only with the ultrasound. The same set up was also used to extract, through power Doppler analysis, 3D palm vein patterns [88], which provide improved distinctiveness with respect to 2D vein patterns achievable with classical techniques (see Figure 5).

An improvement of the acquisition system was presented in Refs. [89,90]; the main difference with the previous set up was the use of a different ultrasonic scanner [91] to replace the previous one, which had shown a too long acquisition time. With the new system, the authors established a database and defined 3D feature extraction procedures based on both palm curvature [92] and line depth analysis [93,94,95]. Figure 6a shows a 3D render of a collected palm where the curvature can be appreciated and Figure 6b displays a 3D template that accounts for lines’ depth. Verification and identification experiments demonstrated the capability of ultrasound systems to provide recognition rates comparable or even better than optical ones.

The possibility to achieve a multimodal system by extracting 3D vein patterns from the same acquired volume, avoiding an additional Doppler analysis, was shown as well [96]. The improved system was exploited also to acquire a wider region by performing multiple scans up to include the whole hand [97,98], and to acquire the 3D palmprint by using a mechanically tilted linear probe [99]. A more reliable system that exploits gel as a coupling medium was successively set up and experimentally evaluated with a homemade database and dedicated recognition procedures [100,101].

## 4. Biometric Systems Based on Impediography

First ultrasound fingerprint images achieved with impediography method were collected at CrossMatch Technologies laboratories, FL, USA [102,103]. The developed sensor was a 64 × 64 pixels matrix with a 126 μm pitch and allows to achieve a resolution of about 250 dpi. It was based on a 1–3 piezocomposite material, i.e., piezoelectric pillars resonating at about 8 MHz embedded in epoxy material. By driving each pillar at its series resonance frequency $f_{s}$, the load in contact with that pillar is detected by measuring its electrical impedance. Due to the high number of elements, it is not possible to interconnect each of them to the driving electronics. Therefore, all the upper electrodes of each column and all the lower electrodes of each row are interconnected, respectively. In this way, a single element can be accessed by selecting the corresponding upper column and lower row, and the electric impedance measured to distinguish between valley (air) and ridge (tissue/water) as in Figure 2.

An accurate study of the device, based on Finite Element analysis, was carried out as well. Several design parameters were evaluated and optimized, including length/width pillar aspect ratio, which was found to be equal to four (length = 160 μm and width = 40 μm) in order to minimize later vibration modes. These modes are due to elastic coupling and generate damping losses [104]. Later on, improved versions of fingerprint sensors were presented at Sonavation inc., a spin-off from Cross-Match since 2006. A swipe sensor, i.e., a device where the images is collected while the finger is shifting on the sensor, composed of 8 × 96 pixel and able to provide a resolution of 350 DPI, was released in market in 2009 [105]; results achieved from another swipe sensor were presented in Ref. [106]. In these papers, results obtained with touch sensors up to 192 × 128 pixels, active sensing area of 12.8 × 19.5 mm${}^{2}$ and resolution of 500 dpi were shown as well. Paper [105] also presents an accurate finite element analysis that investigates several issues like dynamic range (contrast between response in air and water), resolution, linearity of the response, and influence of manufacturing on sensor’s performances.

The possibility to achieve fingerprint images through the impediography method by using a sensor based on CMUT technology has been experimented as well [107,108]. Two approaches were explored: in the first one, the sensor is in direct contact with the finger, while in the second one a polydimethylsiloxane (PDMS) waveguide is interposed between sensor and finger. Preliminary images obtained with both methods are presented; however, image resolution is still quite far away from standard requirements.

Also, impediography technique has been experimented by exploiting aluminum nitride (AlN) piezoelectric thin films working at frequencies higher than 1 GHz [109]. Images of fingerprint rubber phantoms were collected by swiping the phantom itself across a 64 element linear array [110,111]. The capability of such transducer of discriminating among several loads has been demonstrated as well. A main challenge for such high frequency devices is the integration with the electronics.

## 5. Other Biometric Systems Based on Ultrasound

A fingerprint imaging system based on the photoacoustic effect has been recently proposed in Ref. [112]. In this system a dry finger touches an acrylic plate, which, on the other side, is immersed in a water tank that also contains an ultrasound transducer working as a receiver. When a light source with wavelength of 532 nm is directed towards the plate, an ultrasonic wave is generated from thermal expansion and is detected by the transducer (V324-SU, Olympus Inc., Tokyo, Japan), which has a 12 mm focal length, 6 mm diameter, 25 MHz center frequency and 34% fractional bandwidth. First fingerprint images achieved with the proposed method have shown a resolution of about 280 μm, which is still insufficient for main applications. Another drawback of the system relies in the strong dependence of the received signal upon changes of the light source.

PA effect has been used for acquiring 3D vascular patterns as well. Probably, the first system able to visualize microvascular networks was proposed in Ref. [113]. As light source the authors used a commercial Q-switched Nd:YAG laser which pumped a dye laser; laser pulses (6 ns pulses of wavelength 584 nm) were then coupled to an optical fiber. As the ultrasonic transducer they used a 48 elements piezocomposite array with center frequency of 30 MHz and a 50% fractional bandwidth. With this system, 3D images of a rat vascular pattern were achieved for diseases detection applications in medical field.

More recently PA technique has been exploited for imaging 3D hand vein patterns for personal identification [114]. The employed system is composed of a Nd:YAG laser, with wavelength of 1064 nm and a pulse repetition rate of 10 Hz, a 128 elements commercial probe (ATL L7-4, center frequency 5 MHz) and 128 channels imaging systems from Verasonics’ Vantage. To validate the system, a preliminary database was established and a dedicated feature extraction procedure was proposed. Experimental recognition results were encouraging and demonstrated that the system has good recognition rate even when hand pose is changed.

Another technique for acquiring fingerprint was experimented by a research group at Optel (Poland) [115,116]. It is based on acoustic holography and exploits phenomena of scattering and spurious waves’ generation, called contact scattering. The system uses 256 transducers, disposed along different angular directions, that send short pulses and receive the echoes. Some fingerprint images with a declared resolution of 0.1 mm were presented and evaluated only in a qualitative way.

## 6. Conclusions

The present work reviewed the scientific literature that deals with transducers or systems based on ultrasound developed for biometric recognition purposes, starting from first pioneering experiments and including preliminary or partial studies. Several transducer technologies and different ultrasound imaging techniques have been experimented to image various biometric characteristics like fingerprint, palmprint, hand vein pattern, and hand geometry. The majority of research activities has been devoted to fingerprint. To image this biometric characteristic, various technologies have been employed: from first single element piezoelectric transducers scanned along two dimensions to 1D and 2D arrays realized with different technologies, including piezocomposite arrays, CMUTs and PMUTs. Pulse-echo and impediography are the techniques mainly used, but papers that experimented with acoustic holography and, more recently, with photoacoustics can be found as well. In some cases, fingerprint sensors have been successfully integrated to signal processing electronics and, nowadays, several companies are producing ultrasonic fingerprint sensors integrated in portable electronic devices like smartphones or tablets [117,118,119]. Despite of this industrial achievement, papers that quantitatively demonstrate the recognition capability of such devices through verification and identification experiments cannot be found in the scientific literature yet. Ultrasound systems able to collect 3D palmprint and hand vein images have been proposed as well and experimental results carried out on preliminary databases have shown that they are able to provide very good recognition rates. Overall, these research activities demonstrate that, due to its intrinsic properties of enabling 3D images and of being virtually un-spoofed, ultrasound may compete with other technologies in the expanding market of biometrics, which is pervading not only civil applications but also healthcare, automotive, and industrial sectors.

## Figures and Tables

**Figure 1 sensors-19-02317-f001:**
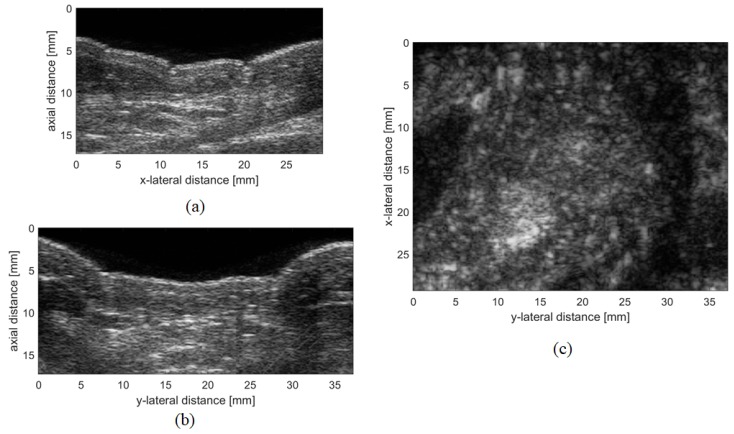
Example of simultaneous renderings of 2D images orthogonal to the three axis extracted from a 3D ultrasound image: (**a**,**b**) brightness (B)-mode images, and (**c**) C-mode image.

**Figure 2 sensors-19-02317-f002:**
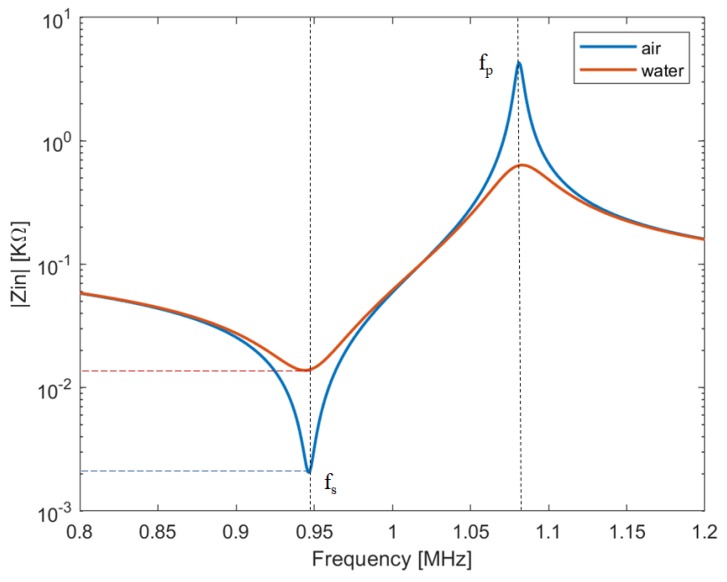
Simulation of the electrical impedance of a piezoceramic element when loaded with air or water to illustrate the impediography technique. The lower the acoustical impedance of the load, the lower the value of the impedance at the series resonance frequency $f_{s}$.

**Figure 3 sensors-19-02317-f003:**
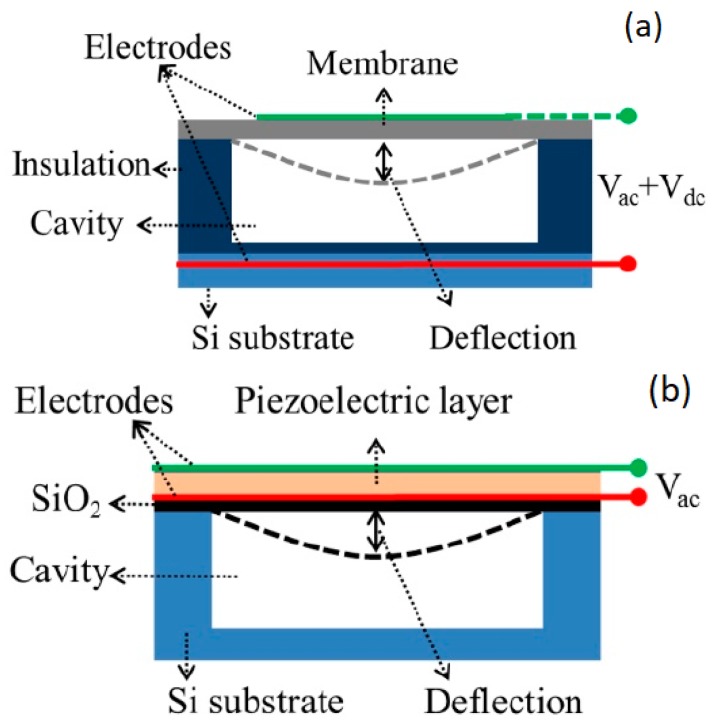
Schematic structures of micromachined ultrasound transducers: (**a**) capacitive micromachined ultrasonic transducer (CMUT) cell: the flexural vibration of the membrane is generated by a bias voltage and an alternate signal; (**b**) piezoelectric micromachined ultrasonic transducers (PMUT) cell: the membrane is composed of at least one piezoelectric layer and one passive elastic layer and the flexural motion is generated by applying only an alternating voltage [55].

**Figure 4 sensors-19-02317-f004:**
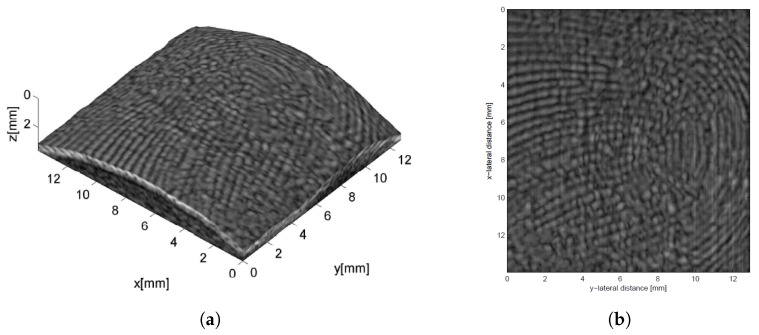
A fingerprint collected by scanning a linear CMUT array along the elevation direction: (**a**) 3D voxel representation, (**b**) 2D image extracted at an under-skin depth of 0.2 mm. In both images, sweat pores are clearly visible. Reprinted with permission from [68].

**Figure 5 sensors-19-02317-f005:**
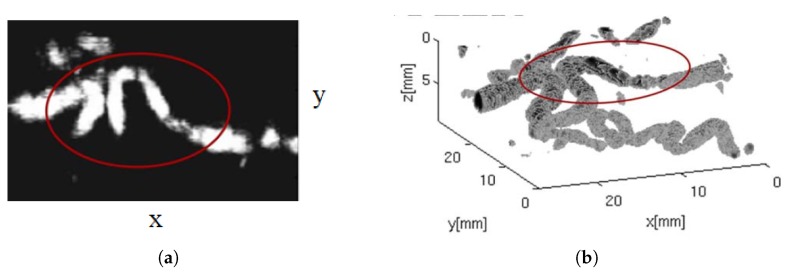
Example of a palm vein pattern collected through power Doppler analysis: (**a**) 2D pattern and (**b**) 3D pattern, which provides improved distinctiveness. Reprinted with permission from [88].

**Figure 6 sensors-19-02317-f006:**
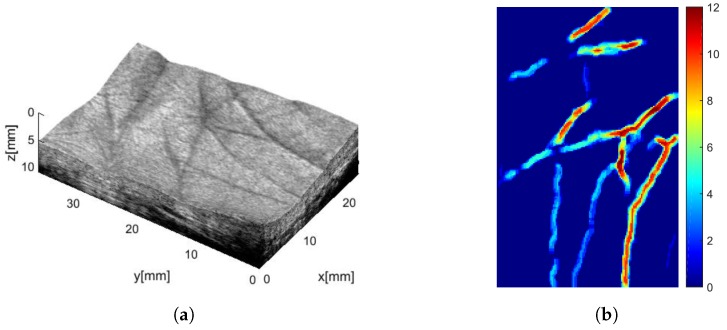
3D palmprint: (**a**) a 3D render of a human palm where palm curvature can be appreciated and (**b**) 3D template accounting for lines’ depth. Reprinted with permission from [95].

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
