# Peer review of "Ultrasound Systems for Biometric Recognition"

_sensors, 2019, doi:10.3390/s19102317_

Reviewer 1 Report

Contained in the attached file

Author Response

The paper is described as an ‘Article’ but is essentially a review of ultrasound technologies for

biometric applications and contains no new information. Therefore, in my view, it should be

considered for publication as a ‘Review’ rather than an ‘Article’.

Yes, it is a “Review” paper. We set it in the latex template.

In the conclusion the paper claims to present an ‘…start of the art…’ review of the techniques: whilst the final sections of the paper do that (sections 2.2.2 onwards, much of the early sections of the paper describe standard ultrasound techniques available in many textbooks (e.g. figures 3 and 4) and should be summarised in a few sentences.

The aim of the present paper is to review all (at our best knowledge) scientific literature that deals with transducers or systems based on ultrasound developed for biometric recognition purposes, starting from first pioneering experiments and including preliminary or partial studies.

Section 2 briefly reports the basic concepts and definitions on Ultrasound imaging techniques (section 2.1) and Transducer technologies (section 2.2) that are employed in the various systems described in the following sections to make more fluid the reading of these last sections. Basic concepts on Biometrics are given in the introduction instead.

Indeed, Figure 3 and 4 describe a (well-known) transducer technology and basic CMUT and PMUT physic working principles, respectively, not ultrasound imaging techniques.

Anyway, in the revised version of the paper, eliminated some sentences on ultrasound technique and also Figure 3 in order to streamline the section.

The review focuses on the sensor and beam characteristics rather than on how these limit the ability to

detect the biometric features they are describing. Similarly, there is limited physics in the sections on transducers which describe their operation and the current research in the area to improve their performance. What would be interesting in both these areas would be to critically review the latest research findings and how these contribute to robust identification.

The review was conceived to provide a description of the method/technique used for collecting the image of the characteristic, the employed technology and the main results achieved, for each research group activity, without go into details, in order to have a more fluent and readable presentation. On the other hand, detailed references are given for each issue, so that the reader can look into specific arguments of interest.

Anyway, in the new version of the paper, we added some critical comments that highlight merits and drawback to the most recent works and improved the conclusions.

Minor and textual issues

Line 39 The stability of vascular patterns depends on depth; surface vascular patterns in the hands

do change over long periods of time and a reference to support the statement is required.

We refer to hand vascular patterns. We clarify this issue in revised version and add a reference.

Line 42 Biometry is not looking at ‘…human organs…’ but surface and near-surface feature and

this should be made clear.

We substituted the word “organs” with “characteristics”, which is more general. However, biometry can be defined as “measurement of living tissue or bodily structures”; hand vein pattern is one example, but also the measurement of inner anatomic elements could be exploited for biometric purposes as shown in a couple of works reviewed here. In these cases, Ultrasound is probably the only technology able to image them.

Line 124 It would be better to say ‘…is used to probe…’ rather than ‘…invests…’

We rearranged the sentence.

Line 258 The sentence beginning ‘It is also to mention…’ should changed to be ‘A finite element methods study has investigated the feasibility…’

We rearranged the sentence as suggested by the reviewer

Reviewer 2 Report

The content of the manuscript is very relevant and properly drafted in a very appropriate way. I recommend the authors to do some major revisions before the manuscript can be accepted for publication.

English language requires a lot of improvement. There are multiple errors in grammar, sentence formation, words usage, spelling and punctuation which makes it really difficult to follow the paper and appreciate it. I recommend the authors to ask a native English speaker to proof read the manuscript.

The title of the manuscript is slightly misleading. The title says, "Ultrasound systems for biometric applications". The manuscript talks in detail about different types of ultrasound systems and talks very less about the biometric applications of it. I suggest the authors to provide a more appropriate title.

In the manuscript the figures, the caption needs to be a little more in detail to convey more details about the various images. Additionally, the X and Y axis of fig 1 is hardly decipherable. Please insert a high quality image.

The introduction and conclusion needs to be improved.

Author Response

The content of the manuscript is very relevant and properly drafted in a very appropriate way. I recommend the authors to do some major revisions before the manuscript can be accepted for publication.

English language requires a lot of improvement. There are multiple errors in grammar, sentence formation, words usage, spelling and punctuation which makes it really difficult to follow the paper and appreciate it. I recommend the authors to ask a native English speaker to proof read the manuscript.

A more accurate English check has been made in the revised version of the paper.

The title of the manuscript is slightly misleading. The title says, "Ultrasound systems for biometric applications". The manuscript talks in detail about different types of ultrasound systems and talks very less about the biometric applications of it. I suggest the authors to provide a more appropriate title.

We substituted the word “applications” with “recognition”. The title is now:  Ultrasound systems for biometric recognition

In the manuscript the figures, the caption needs to be a little more in detail to convey more details about the various images.

We added further details to the captions

Additionally, the X and Y axis of fig 1 is hardly decipherable. Please insert a high quality image.

We did it.

The introduction and conclusion needs to be improved.

We revised both of them, in particular we enhanced the conclusions.

Round  2

Reviewer 1 Report

Acceptable for publication in its current form.

Author Response

We made a further check for grammatical errors.

Reviewer 2 Report

The authors have addressed my concerns adequately. I recommend this manuscript for publication after one more round of proof-reading for grammatical errors

Author Response

(The authors gave the same response as above.)
